# An alternative pathway of enteric PEDV dissemination from nasal cavity to intestinal mucosa in swine

Yuchen Li[1], Qingxin Wu[1], Lulu Huang[1], Chen Yuan[1], Jialu Wang[1] & Qian Yang [1]

Porcine epidemic diarrhea virus (PEDV) has catastrophic impacts on the global pig industry. Although the fecal–oral route is generally accepted, an increased number of reports indicate that airborne transmission may contribute to PEDV outbreak. Here, we show that PEDV could cause typical diarrhea in piglets through a nasal spray. Firstly, PEDV can develop a transient nasal epithelium infection. Subsequently, PEDV-carrying dendritic cells (DCs) allow the virus to be transferred to $CD3^+$ T cells via the virological synapse. Finally, virus-loaded $CD3^+$ T cells reach the intestine through the blood circulation, leading to intestinal infection via cell-to-cell contact. Our study provides evidence for airborne transmission of a gastrointestinal infected coronavirus and illustrates the mechanism of its transport from the entry site to the pathogenic site.

[1] MOE Joint International Research Laboratory of Animal Health and Food Safety, College of Veterinary Medicine, Nanjing Agricultural University, Weigang 1, Nanjing 210095 Jiangsu, PR China. Correspondence and requests for materials should be addressed to Q.Y. (email: zxbyq@njau.edu.cn)

A large-scale outbreak of porcine epidemic diarrhea (PED), characterized by watery diarrhea, dehydration, and vomiting, with up to 90% mortality in suckling piglets, occurred in swine farms in Asia, Europe, and America[1–3]. As the causative agent of PED, porcine epidemic diarrhea virus (PEDV) has spread rapidly between pig farms even over long distances, demonstrating greater transmission potential than other seasonal diarrhea viruses[4,5]. Fecal–oral transmission is believed to be the main mode of PEDV transmission[6]. As many large-scale pig farms in developed countries are equipped with improved disinfection and management systems (sanitation of the pigsty and feed safety are strictly controlled)[7,8], the pathogen is less likely to be spread via the digestive tract (feed or feces). These observations suggest that PEDV may have other routes of infection. Recently, many researchers have highlighted the possible role of airborne transmission of PEDV. Increasing evidence for airborne transmission of gastrointestinal infectious diseases has been reported (e.g., Norwalk viruses and rotavirus)[9,10], but the underlying mechanisms have not been confirmed. Additionally, if PED outbreaks occur in one farm, then neighboring farms are at an increased risk of PEDV infection, and the orientation of PEDV spread follows the prevailing winds in such areas[11,12]. Accordingly, aerosolized particles from pig farms with PEDV outbreak can be identified. As found, PEDV can be detected in all sizes of aerosol particles, with the number of copies of the virus in the particles ranging from $1.3 \times 10^6$ (0.4–0.7 μm) to $3.5 \times 10^8$ RNA copies m$^{-3}$ (9.0–10.0 μm)[13]. In addition, PEDV can survive for a long time (up to 9 months) in the environment and be transmitted over long distances in the field (even 10 miles from pig farms with a PEDV outbreak)[14,15]. Therefore, the cross-talk between PEDV and the respiratory tract merits further attention.

In a recent study, preserved PEDV was detected in the nasal cavity of PEDV-negative piglets in the same room but without contact with piglets inoculated with PEDV[16]. The nasal mucosa is considered a vital gateway for many pathogens (including respiratory and nonrespiratory pathogens). As indicated in previous studies, many dendritic cells (DCs) are widely distributed beneath the nasal mucosa of pigs[17,18] and show a certain degree of susceptibility to PEDV[19].

Submucosal DCs are professional antigen-presenting cells with a potent capacity to capture luminal antigens by forming transepithelial dendrites (TEDs). Such antigen-bearings DCs migrate to the nearby lymph nodes, presenting foreign antigens to T cells and further initiating an effective adaptive immune response[20–22]. Paradoxically, submucosal DCs may sometimes be harnessed by viruses to help them overcome the epithelial barrier, serving as a "Trojan Horse" to evade antiviral immune responses and disseminate into the submucosal layer[23]. The infected DCs then migrate to the nearby lymph nodes and transmit the virus to T lymphocytes (productive or recessive infection)[24,25]. Typically, HIV is a DC-hijacking virus, and DCs might be conducive to its pathogenesis, a mechanism that promotes HIV transmission and infection of CD4$^+$ T cells and further dissemination into the body via the migration of T cells[26–29]. Additionally, DCs are a crucial target of Middle East respiratory syndrome coronavirus (MERS-CoV) replication and a driver of dissemination[30]. Taken together, these data suggest that submucosal DCs in nasal cavity are likely to uptake PEDV and serve as a virus carrier to help PEDV enter and disseminate beyond the nasal mucosa. When DCs are hijacked by the virus, their endocytosis and antigen-processing abilities are limited, and the virus is maintained on dendrites for efficient transfer to T lymphocytes[31]. The virus can then modulate the migratory ability of T cells; sometimes the increased motility of infected T cells facilitates the use of motile cells as drivers to disseminate within and between tissues[29]. Virus-carrying lymphocytes can reach the small intestine and other structures via the blood circulation, penetrate the intestinal mucosa, and transmit the virus to target cells, becoming a vital source of infection[32,33]. The virus-carrying lymphocytes transmit the virus to host cells, which is called transfer infection. For instance, researchers have suggested that varicella-zoster virus (VZV)-infected T cells can enter the gut and establish latency in enteric neurons in vivo[34]. Moreover, PEDV can enter the blood and induce viremia in suckling piglets, which implies that PEDV possesses the ability to colonize some cell types in the peripheral blood[35].

In the present study, the hypothesis for airborne PEDV caused intestine infection is proposed. A PEDV intranasal challenge experiment was performed to verify our hypothesis. A series of coculture models were established in vitro using nasal epithelial cells (NECs), DCs, T lymphocytes, and Vero cells to gain more insights into the mechanism by which PEDV enters the intestinal epithelium via the nasal cavity. This study is the first to demonstrate evidence of the airborne transmission of a gastrointestinal infected coronavirus. Moreover, pigs are closely related to humans in terms of anatomy, genetics, and physiology[36]. Thus, the use of pigs as an animal model may help to reveal the pathogenicity of human viruses that possess similar pathogenic characteristics to PEDV.

## Results

**PEDV intranasal inoculation causes typical PED symptoms**. To detect whether the PEDV can infect piglets through intranasal inoculation, we carried out animal challenge experiments. Piglets were assigned to 3 groups (I: Control, II: PEDV intranasal inoculation, and III: PEDV oral inoculation). Clinical signs included severe watery diarrhea with vomiting that were first detected in oral PEDV-inoculated piglets at 46 h post-infection (hpi). At 60 hpi, the intranasal inoculated piglets from group II began to exhibit classical PEDV symptoms, including acute, severe watery diarrhea, depression, and lethargy. Abundant yellow, foul-smelling watery stools were also observed around the perianal region of the piglets (Fig. 1a). Then, the piglets were anesthetized with pentobarbital sodium (100 mg kg$^{-1}$) and sacrificed for macroscopic examination at 66 hpi. After intranasal inoculation, all piglets exhibited moderately thin and transparent intestinal walls in the small intestine, with an accumulation of large amounts of fluid in the intestinal lumen (Fig. 1a). Pathological examination of the intestine of piglets with diarrhea after intranasal inoculation revealed multifocal to diffuse villous atrophy (Fig. 1b). Immunofluorescence analysis (IFA) showed a large amount of PEDV-positive cells in the small intestine of piglets with diarrhea, and PEDV antigens were mainly observed in the cytoplasm of villus epithelial cells (Fig. 1b). Quantitative reverse transcription-polymerase chain reaction (qRT-PCR) showed viral RNA expression in different tissues of the piglets after intranasal inoculation. PEDV mainly colonized the jejunum and ileum, and peak viral RNA titers reached up to 5.19 log10 in the jejunum, this titer was significantly higher than that in the other tissues (Fig. 1c). PEDV also showed a broad tissue tropism, and low-level RNA expression was detected in the epithelium of nasal and lung tissues (Fig. 1c). Western blot results further validated the PEDV level in different tissues, and a significant quantity of PEDV N protein was detected in the jejunum and ileum. However, no immunoreactivity was observed with proteins of the nasal epithelium or lung despite the low level of RNA expression (Fig. 1d).

**Replication of PEDV in the nasal cavity of piglets**. After nasal inoculation, the virus was first detected in the nasal mucosa, trachea, and lung, while no viral RNA was detected in the gastrointestinal tissue (Fig. 2a). The virus showed a certain degree of

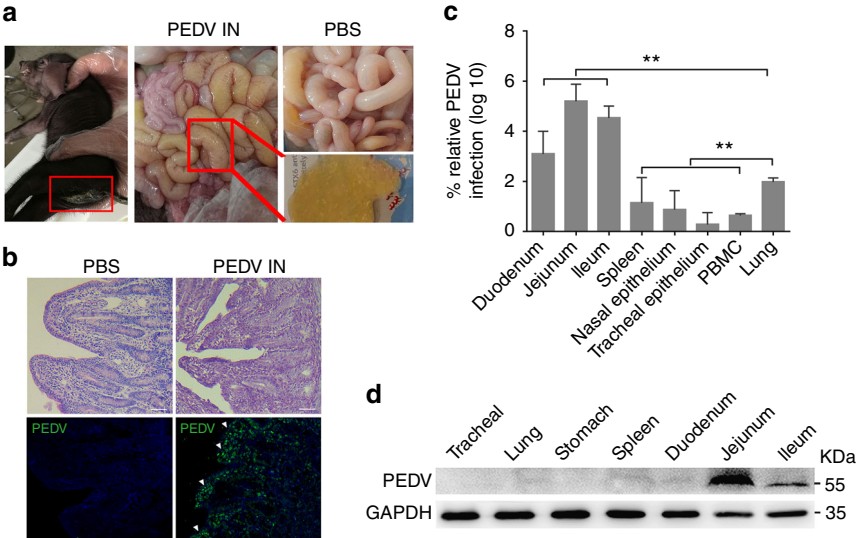

**Fig. 1** PEDV causes typical PED symptoms after intranasal inoculation. **a** Acute watery diarrhea and gross lesions of the intestine in piglets after PEDV intranasal inoculation at 66 hpi. **b** Hematoxylin and eosin (H&E) staining and IFA of the intestine of an intranasal inoculated pig at 66 hpi. PEDV antigen was located in the atrophied intestinal villus (white arrowhead). IN: intranasal inoculation. The scale bar represents 50 μm. Blue, DAPI; green, PEDV. **c** Viral RNA expression in different tissues of the diarrheic piglets after intranasal inoculation, $n = 6$ from 3 piglets per group. Data are the mean ± SD. Statistical significance was using one-way ANOVA. **P < 0.01. **d** Protein expression of PEDV in different tissues of diarrheic piglets as determined by Western blotting with a mouse mAb against N protein. At least three independent experiments were performed

replication in the nasal epithelium within 24 h, with RNA titers reaching a peak of 3.14 log10 within 12 h and declining at 24 hpi. At 24 h, many PEDV gene copies were detected in the jejunum and ileum at higher levels than those in other tissues. The nasal mucosa of piglets was isolated at different time points after intranasal infection to further investigate the replication features of PEDV in NECs. Similar to the PEDV RNA expression in the nasal epithelium, fluorescence-activated cell sorting (FACS) analyses showed that PEDV colonized NECs, with 0.79% positive cells observed at 3 h, and maintained the infection between 12 and 24 h (PEDV-positive cells ranging from 1.81 to 2.02%). In contrast, the level of virus in NECs declined at 66 hpi (Fig. 2b, c). Moreover, after infection, the nasal cavity was fixed after infection for 12 h and subjected to IHC analysis. Virus-positive cells were unevenly distributed throughout the cytoplasm of NECs in regions I–IV (Fig. 2d, Supplementary Fig. 1c). In contrast, section IV in which most DCs and other lymphocytes accumulated, showed an increased proportion of PEDV-positive cells (Fig. 2d, e).

**Nasal epithelium culture and susceptibility to PEDV**. We used a nasal epithelial cell model (NECM) (Fig. 3a) to verify the consequences of PEDV infection in vitro. Morphological examination of NECs cultures at 6 days revealed successful establishment of the NECM model (Fig. 3b). The stripped nasal mucosa is shown in Supplementary Fig. 2a, and the cilia structure in some of the isolated nasal epithelium was preserved (Fig. 3b). Air–liquid interface establishment, tight junction formation, epithelial cell marker staining, and transepithelial electric resistance changing (TEER) indicated that the NECs were well polarized (Fig. 3b, Supplementary Fig. 2b, c, f). Examination of the NECM using a scanning electron microscope (SEM) further validated the air–liquid interface formation and revealed that these cultures were composed of different epithelial cell subtypes, including ciliated and nonciliated cells (Fig. 3b). FACS analysis demonstrated that NECM was devoid of fibroblast contaminants

(Supplementary Fig. 2e). Porcine aminopeptidase N (pAPN) is a functional receptor for PEDV. The distribution of pAPN in NECM was shown in Supplementary Fig. 2d.

No apparent visual cytopathic effect (CPE) was observed in NECs (unpolarized) and the NECM after 48 h of PEDV infection (Fig. 3c). Infectious virus was recovered only from the apical side of the NECM, not at the basolateral side of the NECM (Fig. 3d, e). Additionally, compared with the NECM, unpolarized NECs showed high susceptibility to PEDV within 48 h, with a peak viral titer of 3.4 log10 PFU ml$^{-1}$ at 24 h (Fig. 3e).

**Luminal PEDV captured by DCs in vitro and in vivo**. The nasal mucosa can be considered a vital gateway for many pathogens. After 12 h post intranasal administration of PEDV, the capture of luminal PEDV by DCs was detected in the nasal-associated lymphoid tissue (NALT) and lamina propria of the nasal mucosa in vivo (Fig. 4a, b). We speculated PEDV might exploit submucosal DCs in the nasal cavity to enter and disseminate beyond the nasal mucosa. A NECM/DCs coculture system was used to study virus uptake by DCs (Supplementary Fig. 3b). Of note, DCs coculture or PEDV challenge did not modify the morphological characteristics of tight junctions (Supplementary Fig. 3c). Moreover, the TEER of the NECM/DCs coculture system was not influenced by DCs or PEDV. Barrier disruption was performed by the addition of EDTA to the apical medium (Supplementary Fig. 3d). The apical side of the NECM/DCs coculture system was inoculated with fluorescence-labeled PEDV (Fig. 4c). In the coculture system, inoculation with live PEDV resulted in a greater number of PEDV-loaded submucosal DCs than did inoculation with inactivated virus (Fig. 4d, e). The number of virus-loaded submucosal DCs markedly increased with the increased PEDV inoculation dose (Fig. 4f). To explain how PEDV entered the DCs in the coculture system, we used an antibody against histocompatibility complex class II (MHCII) to identify DCs in z-orthogonal views using confocal laser scanning microscopy (CLSM) (Fig. 4g). DCs were clearly visualized as enlarged TEDs

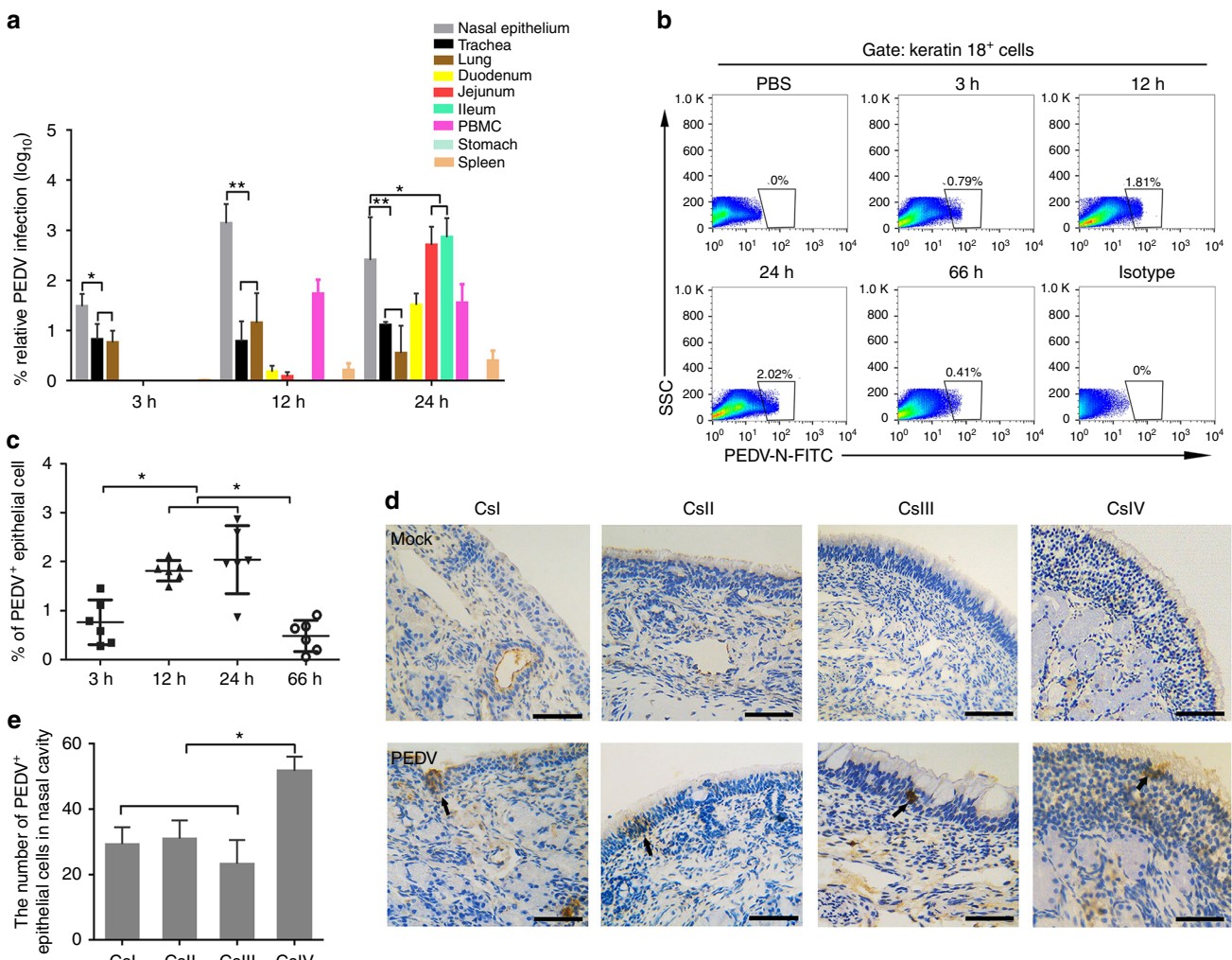

**Fig. 2** Replication and distribution of PEDV in piglets' nasal cavity. **a** RNA expression levels of PEDV in different tissues of piglets at different time points after intranasal PEDV challenge, $n = 6$ from 3 piglets per group. **b** For FACS analyses, pigs were nasally administered PEDV at indicated times. Then, individual cells isolated from the nasal mucosa (both left and right nasal cavity) were gated based on CK18+ (marker of epithelial cells), and viral infection was detected by PEDV N protein staining, $n = 6$ from 3 piglets per group. **c** Quantification of the FACS results as shown in (**b**). **d**, **e** IHC results showing the distribution pattern of PEDV in four cross-sections (I, II, III, and IV) of the piglet nasal cavity at 12 h post intranasal infection. The numbers of PEDV-positive cells (black arrowheads) in different part of the cavity were counted in six random fields (40×) from three cross-sections. The scale bar represents 100 μm. All data shown are the mean results ± SD from three independent experiments. Statistical significance was using one-way ANOVA. NS no significance, *$P$ < 0.05, **$P$ < 0.01

across the tight junctions of NECs and had internalized luminal PEDV, while DCs in the control group were mostly beneath the epithelial cells (Fig. 4h). Transmission electron microscopy (TEM) observations of the NECM/DCs coculture system further confirmed that DCs extended dendrites into the filter and captured PEDV (Fig. 4i). Integral PEDV virions could be found in the extended dendrites and cytoplasm of DCs on the basolateral side (Fig. 4i).

**CCL25 and NF-κB pathway are involved in DCs recruitment.** DCs maturation is a key step in the subsequent antigen-specific immune response. In the coculture system, although the levels of MHCII stimulated by PEDV were not different from those of the control, the increases in swine workshop cluster 3a (SWC3a) and cluster of differentiation 1a (CD1a) after PEDV inoculation were virus dose-dependent, demonstrating the maturation of submucosal DCs (Fig. 5a, b). To better understand the molecular cues required for DCs recruitment and TED formation in the nasal passage, we assessed the expression of many chemokines

associated with DCs migration. Chemokine C–C motif ligand 25 (CCL25) was upregulated at both RNA and protein levels after PEDV inoculation (Fig. 5c, d). Additionally, after PEDV inoculation, an increase in P65 in the nucleus and p-P65 in the cytoplasm confirmed the activation of nuclear factor kappa B (NF-κB) in NECs (Fig. 5e). However, pretreatment of the NECM with an inhibitor of NF-κB (BAY 11-7082), but not with the DMSO control significantly reduced the number of PEDV-containing DCs and TEDs on the apical side in the coculture system (Fig. 5f–h).

**T cells acquire PEDV from DCs and enter the peripheral blood.** PEDV was detected in CD3+ T cells in peripheral blood mononuclear cells (PBMCs) of piglets within 24 h after intranasal inoculation. The proportion of virus-loaded CD3+ T cells increased at an early stage and peaked with 4.15% at 12 h (Fig. 6a, b). Virus in CD3+ cells may be acquired from DCs, which take up virus into the lumen across the nasal mucosa, migrate, and engage with T cells. Coculture of DCs and T cells was employed to study

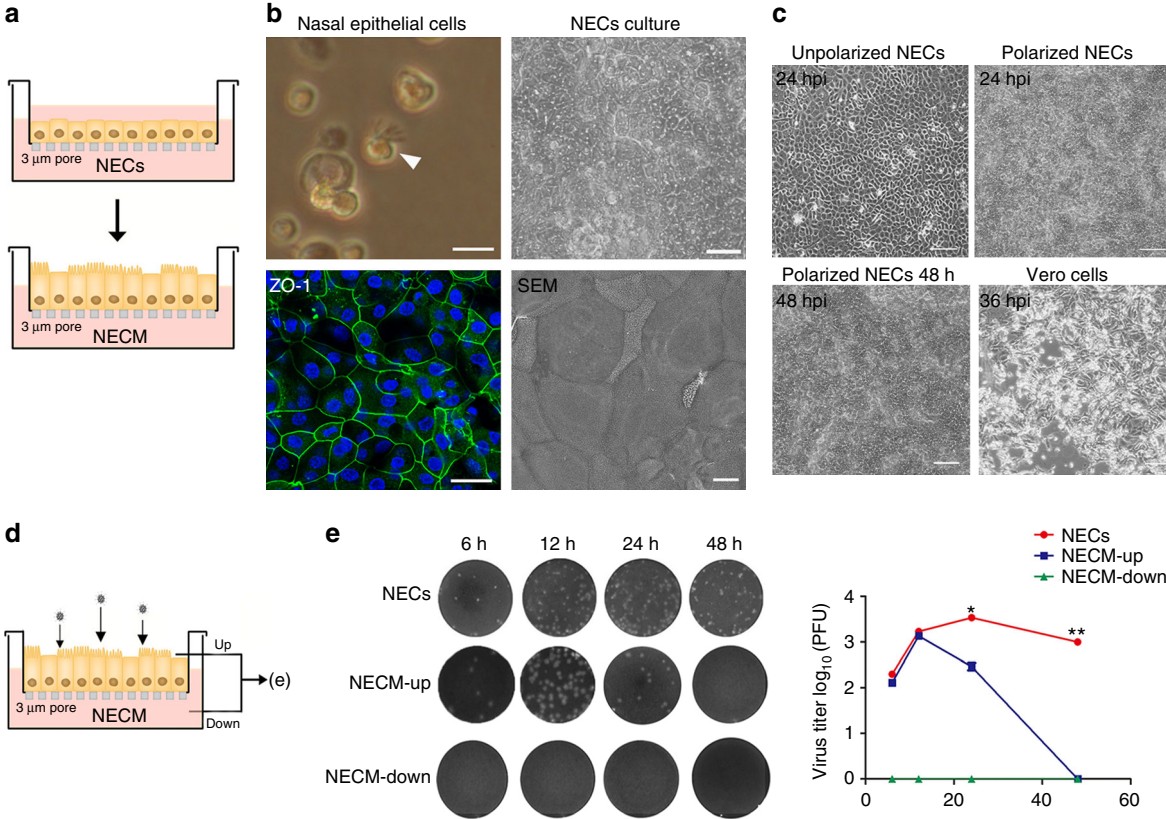

**Fig. 3** NECM establishment and susceptibility to PEDV evaluation. **a** Schematic of differentiation and polarization of porcine NECs after seeding onto collagen-coated porous inserts. **b** Ciliated epithelial cells and the air–liquid interface after 6 days of NECs culture were observed by phase contrast microscopy. Expression of the tight and adherent junction marker protein ZO-1 (green) and a SEM image demonstrating the polarization of NECs. Cell nuclei were stained with DAPI (blue). **c** The PEDV replication in unpolarized NECs (MOI) = 1, polarized NECs cultures (MOI) = 1, and Vero cells (MOI) = 0.1 were observed by light microscopy. Bars, 20 μm. **d** The NECM and NECs were infected with PEDV (MOI 0.1) through the apical membrane. Media from both apical and basolateral chambers of the NECM were collected at various time points. **e** The viral titer was determined by a plaque assay and quantified. The data shown are the mean results ± SD from three independent experiments. The comparisons were performed with t-tests (two groups) or analysis of variance (ANOVA) (multiple groups). *$P < 0.05$, **$P < 0.01$

PEDV transmission from DCs to T cells. FACS analysis showed that PEDV could be transferred from DCs to CD3$^+$ T cells, with 2.64 and 5.33% PEDV$^+$ CD3$^+$ T cells detected at 2 and 12 h of coculture, respectively (Fig. 6c, d). In contrast, the PEDV RNA expression level in T cells remained stable within 48 h, inconsistent with the FACS results (Fig. 6e).

To detect whether PEDV could induce the conjugate structure between DCs and CD3$^+$ T cells, we performed a conjugate formation experiment. FACS analysis of conjugate formation revealed that 42.8% of PEDV-treated DCs formed conjugates with T cells, while DCs in the control group and PEDV (inactivated)-treated group showed reduced conjugate formation compared with those in the live virus treatment group (9–15% of PBS or PEDV (inactivated)-treated DCs formed conjugates with T cells) (Fig. 6f). Furthermore, DC/T cell conjugates induced by PEDV were observed by CLSM, which revealed virus in the conjugate structures formed by the two cells (Fig. 6g). TEM imaging also showed that PEDV was taken up by DCs located within large compartments (Fig. 6h, arrowheads) at the synapse, a cell–cell contact that involves extensive interdigitation (black frame) of the respective cell membranes (Fig. 6h). The combination of the membrane encasement, deep virion channels and interdigitation between the donor and target cell membranes served to ensure that PEDV transfer to T cells occurs in a highly secluded environment. The panel II of Fig. 6h shows that PEDV was successfully transferred from DCs to T cells, and integrated

coronavirus-like particles (arrowheads) were present in the cytoplasm of T cells.

**Virus-loaded T cells can cause intestine infection**. Next, to evaluate whether PEDV-loaded T cells could reach the small intestine, we performed a competitive T cell autotransfusion assay in piglets. After 24 h, more CFSE-labeled T cells (coculture with PEDV-treated DCs) entered the blood circulation and jejunum than did CM-Dil-labeled T cells (coculture with mock infection DCs) (Fig. 7a, b). Observations of frozen sections of the jejunum further confirmed the FACS results (Fig. 7c). We further assessed whether the transferred PEDV-loaded T cells could transmit the virus to host cells. T cells (donor cells) acquired the virus from DCs and were sorted and cocultured with PEDV-susceptible Vero cells (recipient cells) in two forms, contact or noncontact (Fig. 7d). Among CD3$^+$ T cells, 3% were PEDV-positive in both the contact and noncontact groups at the original stage. Surprisingly, PEDV-positive T cells again appeared in both groups after 96 h of coculture. Particularly in the contact group, as much as 51.9% of PEDV$^+$ CD3$^+$ T cells were detected, which was much higher than those in the direct contact group (Fig. 7e). As shown in Fig. 7f, the absence of virus in the supernatant at 2 h eliminated the possibility of residual PEDV outside the T cells in the medium. In contrast to Vero cells in the noncontact group, Vero cells in the contact group with T cells exhibited higher PEDV RNA expression levels and

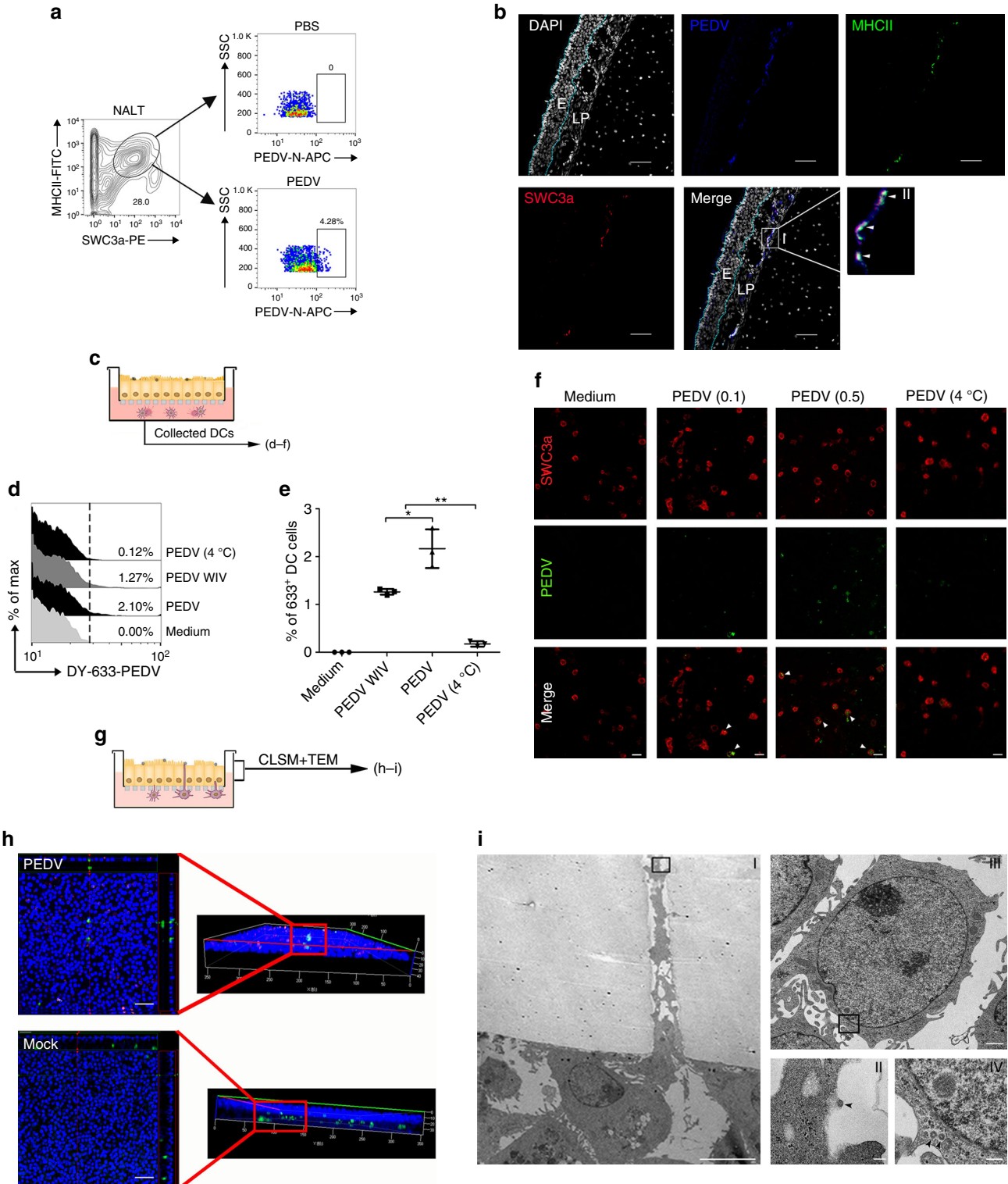

virus titers in the medium (Fig. 7f, g). An apparent CPE appeared in Vero cells in the contact group at 96 h, while a mild lesion was observed in Vero cells in the noncontact group (Fig. 7h). Moreover, in the contact group, we observed many Vero cells displaying dramatic changes in cellular morphology, with significant increases in membrane extensions in which the filopodia appeared to be very thick, essentially forming a large branch emanating from the central body of the cell and contacting T cells (Fig. 7i).

The above donor T cells and target Vero cells were further analyzed by TEM (Fig. 7j). The variation in morphological characteristics, such as an elongated cell and cellular pseudopodium, were observed in T cells compared with mock-infected T cells (coculture with PBS-treated DCs). Moreover, the accumulation of a larger number of PEDV particles in the cytoplasm of target Vero cells (white asterisk) after 96 h by contact culture validated the transfer of infection between donor and target cells.

**Fig. 4** Capture of luminal PEDV by lamina propria (LP) DCs. **a** At 12 hpi, virus-loaded DCs in NALTs were analyzed by FACS. DCs cells were gated based on FITC-labeled MHCII and PE-labeled SWC3a, viral uptake by DCs in piglets were further detected by APC-labeled PEDV N mAb. SSC: side scatter. **b** Moreover, IV sections of the nasal cavity were detected by IFA. DCs located in nasal mucosa were immunolabeled with anti-SWC3a mAb (red) and anti-MHCII mAb (green). Colonization of PEDV was detected by pig anti-PEDV polyclonal antibody (blue); enlarged images of the corresponding region in the white box in panel I (**b**) was magnified in panel II. Bars, 20 μm. **c** Schematic of the experimental setting used to study viral uptake in the coculture system. **d** DCs were collected from the coculture system at 1 hpi and detected by FACS. **e** The quantified FACS results are shown in (**d**), data shown are the mean results ± SD, $n = 3$ per group. Statistical significance was using one-way ANOVA. *$P < 0.05$, **$P < 0.01$. **f** Uptake of PEDV by submucosal DCs was also determined using IFA. DyLight 633-labeled PEDV (green) was observed within DCs (SWC3a+, red). Bars, 5 μm. **g** Schematic of the experimental setting used to study viral capture by DCs in the coculture system. **h** The filters were processed for CLSM after DyLight 633-labeled PEDV inoculation. In the 3D views, the TEDs of DCs (MHCII, green) were crossing the nasal epithelial cell monolayer (DAPI, blue). Internalization of viruses by TEDs was also observed (red frame) and enlarged in cross-sectional images. In contrast, basolateral DCs showed no response when inoculated with medium. Bars, 20 μm. **i** After 1 h of PEDV incubation, the filters from the coculture system were processed for TEM. An enlarged image of the region in the black frame in panels I and III was shown in panels II and IV, respectively. PEDV virions were marked with black arrowheads in panels II and IV. Bars, 1 μm (I); 0.5 μm (III); and 200 nm (II and IV). All results are representative of three independent experiments

## Discussion

Due to its devastating impact on the global pig industry in the US, PEDV has aroused vast concern worldwide[1,2,6,37]. Most published research regarding PEDV has focused on its epidemic characteristics, structural features, and vaccine development; fewer studies have been conducted to delineate its mechanisms of pathogenic pathways, which are generally thought to occur via the digestive tract[38]. However, based on its previously reported airborne characteristics, we propose that another pathogenic pathway may be involved in PEDV infection in piglets[13,14]. In this study, we demonstrate an alternative pathway of PEDV infection, in which PEDV can disseminate from the entry site (nasal epithelium) to the pathogenic site (intestinal mucosa) by utilizing immune cells.

Through intranasal inoculation, we found that PEDV could cause typical PED symptoms in piglets, and the fecal titers achieved through intranasal inoculation were the same as those achieved through oral inoculation. The longer incubation period observed in the intranasal inoculation group may be attributed to the unique dissemination pathway of the virus, which spreads from the nasal epithelium to the intestine. The nasal mucosa is the first site to be directly exposed to inhaled pathogens, representing an important portal for pathogen entry[39]. In our study, PEDV developed a transient nasal epithelium infection, and more virus was detected at the rear of the nasal cavity. Transient patterns of virus shedding were observed, with viral loads in tissues and host RNA expression peaking early in infection and decreasing over time. We speculate that PEDV primarily affects the nasal epithelium in a transient manner, by which the airborne virus can resume its activity and produce progeny virions in the nasal cavity. Moreover, according to previous research, DCs and many other lymphocytes mainly accumulate at the rear of the nasal cavity[17].

An interaction between the virus and immune cells is frequently observed, and may benefit or impede the virus. DCs are widely distributed in the mucosal lamina propria of various tissues, such as the respiratory, gastrointestinal, and genital tracts. DCs have the ability to take up antigens at these sites, predisposing them to become primary targets for viral infection[27]. Submucosal DCs are characterized by the extension of cellular processes into the lumen for continuous surveillance[40,41]. However, as a double-edged sword, the function of DCs is sometimes exploited by the virus to overcome the mucosal barrier. For instance, HIV-infected DCs are mainly responsible for establishing a foothold of infection[27]. DCs are exploited by VZV invading the mucosa and are known as the immune systems' Achilles heel during VZV infection[25]. Additionally, many other viruses, including Dengue virus, measles virus, and cytomegalovirus, can utilize DCs to enter the body, disseminate in humans, evade antiviral immune responses, and cause disease[42–44]. In our study, DCs formed TEDs to capture luminal PEDV virions in vitro and in piglets, and then helped PEDV overcome the nasal epithelial barrier. Furthermore, integral viral particles may be excluded from degradation by acid proteolysis for more efficient vesicle-mediated transfer to other cells[27]. Moreover, PEDV has evolved strategies to exploit DCs maturation to promote its dissemination, while unchanged levels of MHCII expression may induce interference with MHCII-associated antigen presentation[45,46]. The family of NF-κB transcription factors plays a central role in coordinating the expression of a wide variety of chemokines, a family of small molecules that induce chemotaxis of neutrophils, mononuclear cells, or lymphocytes[47,48]. In our study, activation of NF-κB in NECs by PEDV was associated with the release of the chemokine CCL25 and mobilized DCs to accumulate in the submucosal regions. Taken together, these results imply that, although the virus can be released only from the apical side of the NECM, submucosal DCs may function as an important portal for PEDV entry into the nasal mucosa.

DCs can transfer HIV to CD4+ T cells in which it replicates explosively[27,31]. VZV exploits DCs as "Trojan Horses", allowing it to shuttle to the tonsils and regional lymph nodes, where VZV-infected DCs establish T cell infection[25]. In our study, DCs could transfer PEDV to CD3+ T cells by inducing functional cross-talk within the conjugated structure between DCs and CD3+ T cells, and the virus was detected in these structures. Considering the structural similarity, the large membranous puncta from PEDV-containing DCs to T cells are potentially similar to the previously described virological synapses responsible for the HIV-1 transinfection process during HIV pathogenesis[49]. However, the detailed molecular mechanism by which PEDV is transferred from DCs to T cells has not been defined.

In addition, PEDV-loaded CD3+ T cells can enter the peripheral blood and reach the intestinal mucosa through the blood circulation. Such adventures are replete with difficulties and challenges for free PEDV, such as limited diffusion, anatomical barriers, and soluble immune factors (complement and natural antibody)[50]. Allowing infected T cells to retain robust motility and serve as migratory vehicles might therefore be a viral strategy for efficient local and systemic dissemination[33]. Numerous high endothelial venules (HEV) are distributed in Peyer's Patches and lymph nodes of the intestine, through which virus-carrying T cells can enter the intestinal mucosa after reaching the small intestine through the blood circulation[51].

Subsequently, T cells can become an important source of infection. VZV-infected T cells can potentially transfer VZV to

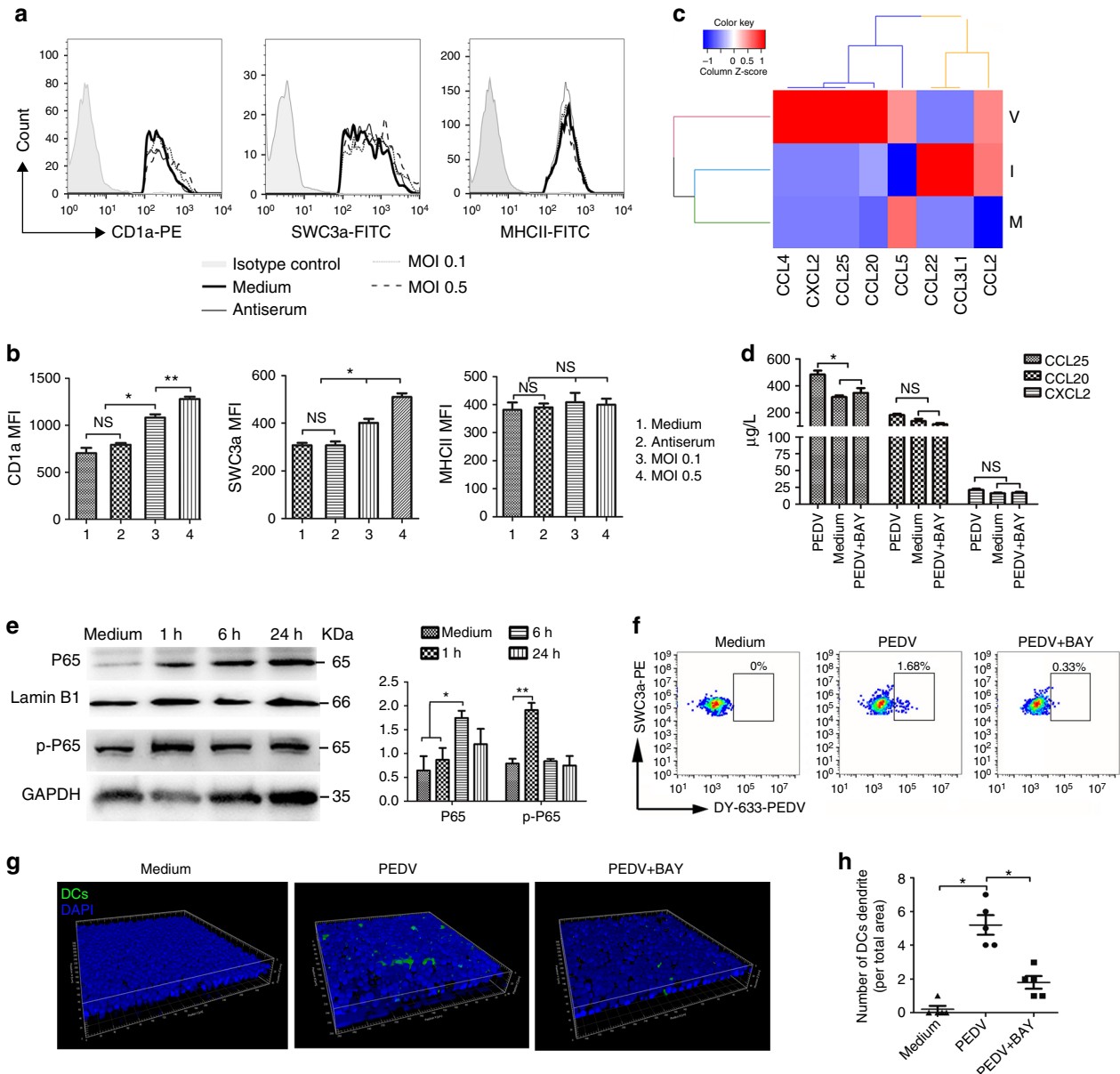

**Fig. 5** Role of CCL25 and NF-κB pathway in DCs recruitment and TED formation. **a**, **b** After incubation with medium, PEDV or PEDV plus antiserum on the apical side of the NECs for 20 h, basolateral DCs were collected. The expression of phenotypic marker including CD1a, SWC3a, and MHCII on DCs was analyzed by FACS. $n = 3$ per group. **c** After 1 h of PEDV incubation, qRT-PCR was used to determine the relative mRNA expression of a series of chemokines related to DCs in NECs, $n = 3$ per group. **d** The protein expression of CCL25, CCL20, and CXCL2 in the medium of the basolateral side were detected using ELISA kits, $n = 3$ per group. **e** After PEDV inoculation, the phosphorylation of p65 and nuclear p65 in the NECs of the coculture system was detected at the indicated time by Western blotting. **f** The coculture system was pretreated with an NF-κB inhibitor for 2 h (DMSO as a negative control) by upper compartment inoculation, followed by inoculation with DyLight 633-labeled PEDV for 1 h. Basolateral DCs were collected and analyzed by FACS. **g** IFA of filters from the coculture system and a three-dimensional (3D) rendering of representative fields showing that basolateral DCs sent dendrites (MHCII, green) to creep through ECs in response to PEDV and/or inhibitor. **h** Quantitative analysis of TEDs was performed. The number of TEDs was counted from five random fields of view at a unit area (0.078 mm²) for each of three individual filters and depicted as a dot plot with each dot representing a field. All data are the mean ± SD, comparisons performed with one-way ANOVA. *$P < 0.05$, **$P < 0.01$. The results are from at least three different experiments

the intestinal tract and establish latency in enteric neurons in vivo[34]. The virus-carrying lymphocyte induction of infection in susceptible cells is known as transfer infection. The specific mechanism of transfer-infection has been more extensively studied in HIV, which can spread from T cells to astrocytes and uninfected T cells via cell-to-cell contact, indicating a unique interpretation of rapid HIV transmission during the early process of infection[52,53]. PEDV infection of Vero cells occurred efficiently by cell-to-cell contact with PEDV-infected CD3+ T cells. CPE of Vero cells in the contact group was apparent at 96 h, while the minor lesion was concomitantly observed in the noncontact group. PEDV could be transferred within 2 h through T cell–Vero contact coculture; in turn, large amounts of virus were transferred from infected Vero cells to T cells. We also detected large extensive membranous structures, either as membrane sheets or filopodial bridges, between Vero and CD3+ T cells, that might be

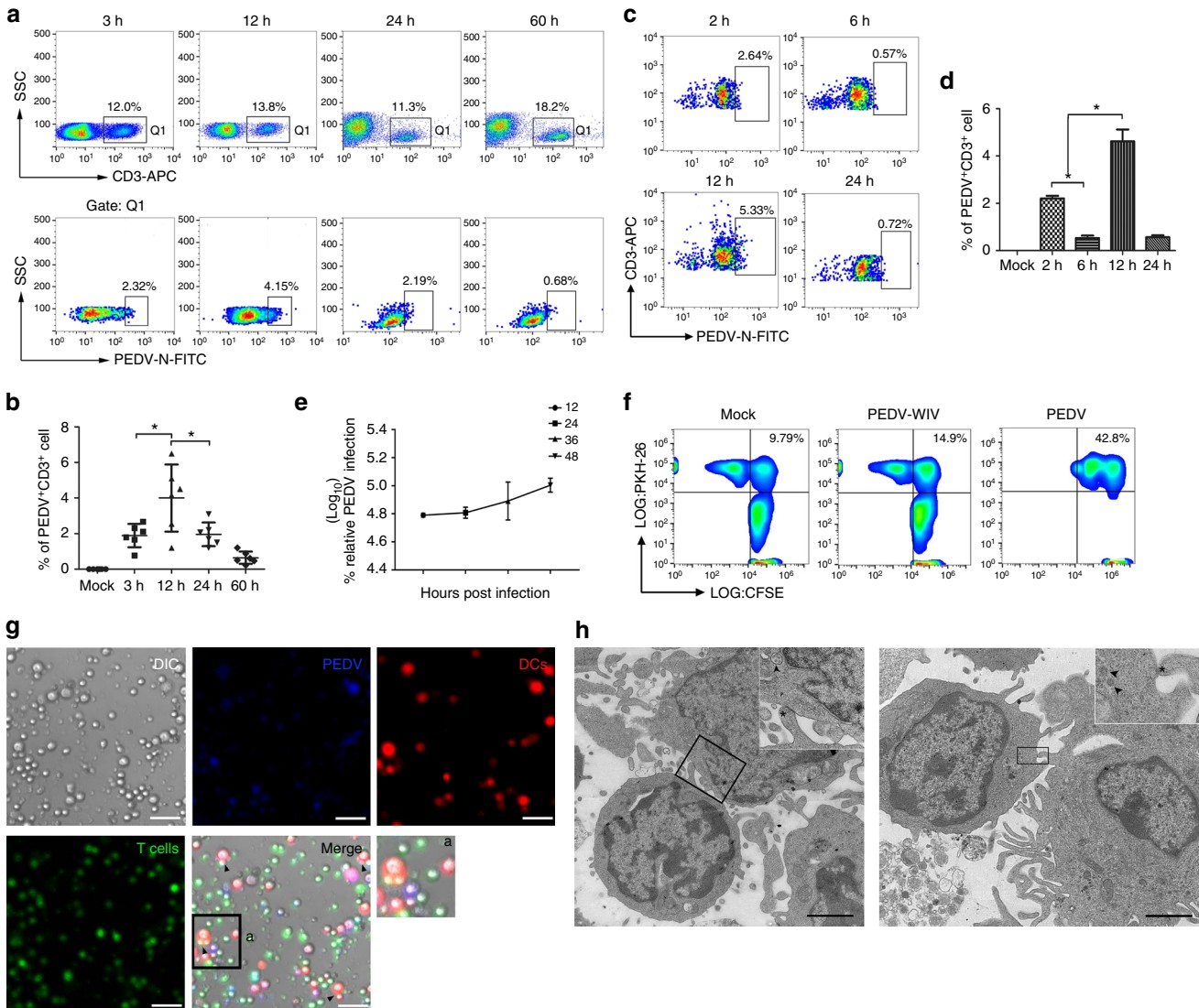

**Fig. 6** CD3+ T cells acquire PEDV from DCs and enter the peripheral blood. **a** After PEDV intranasal inoculation, blood samples were collected at the indicated time during which PBMCs were isolated and analyzed by FACS. The cells were gated based on CD3+ T cells (Q1), and gated cells were further selected based on the uptake of PEDV. **b** Quantification of the FACS results is shown in panel (**a**), $n = 6$ from 3 piglets per group. **c, d** PEDV-pulsed DCs were cocultured with CD3+ T cells to detect the transmission of DC-associated PEDV to CD3+ T cells. Then, PEDV transmission was quantified by analyzing the CD3+ PEDV+ population, $n = 3$ wells per group. **e** Furthermore, after coculturing with CD3+ T cells for 2 h, DCs were separated by magnetic beads separation (MACS). The remaining CD3+ T cells remained in culture, and PEDV RNA expression levels were determined at the indicated time; the results of a representative experiment are shown, $n = 3$ per group. **f** FACS profiles of conjugate formation between CFSE-labeled T cells (x-axis) and PKH26-labeled PEDV-pulsed DCs (y-axis). Conjugates are apparent in the upper right quadrant, and the percentages of cells are shown. **g** Conjugation between CFSE-labeled T cells (green) and PKH26-labeled DCs (red) is indicated by CLSM, and the typical conjugation structure between the two cells is enlarged in panel (**a**). PEDV (blue) was detected in the conjugate structure using an antibody against N protein. Bars, 20 μm. **h** Electron micrographs of DC-T cell conjugates. PEDV-carrying DCs were cocultured with CD3+ T cells for 1 h and collected for TEM analysis. The sites of interaction between DCs and T cells showed firm interactions (black frame). DCs and T cell membranes were closely apposed at the tips of the protrusions (black asterisk). The fine ultrastructure of the virus particles (black arrowheads) was observed in the T cells adjacent to the conjugate structure. Bars, 2 μm. All data are the mean ± SD, comparisons performed with one-way ANOVA. *$P < 0.05$. The results are representative of three independent experiments

involved in PEDV transmission between CD3+ T and Vero cells. Whether these structures are present during PEDV cell-to-cell transfer in piglets or the role they play in PEDV cell-to-cell transmission will require further exploration. Cell-to-cell spread of PEDV in *trans* or in *cis* may permit efficient ongoing replication and spreading despite a variety of defense strategies in the intestinal mucosa[54].

Our study demonstrates an alternative pathogenic pathway of PEDV, which can cause typical pathogenic symptoms in piglets through nasal cavity inoculation. The exact mechanism by which PEDV enters the intestinal epithelium through the respiratory route has been investigated (Fig. 8). The tropism of PEDV for NECs is responsible for virus propagation and accumulation at the rear of the nasal cavity. However, DCs may play an important role in virus entry into the nasal mucosa and transfer of the virus to T cells. Furthermore, the virus-carrying T cells could reach the intestine via the blood and lymph circulation and transfer the virus to the intestinal epithelium. These findings are helpful in facilitating strategies for

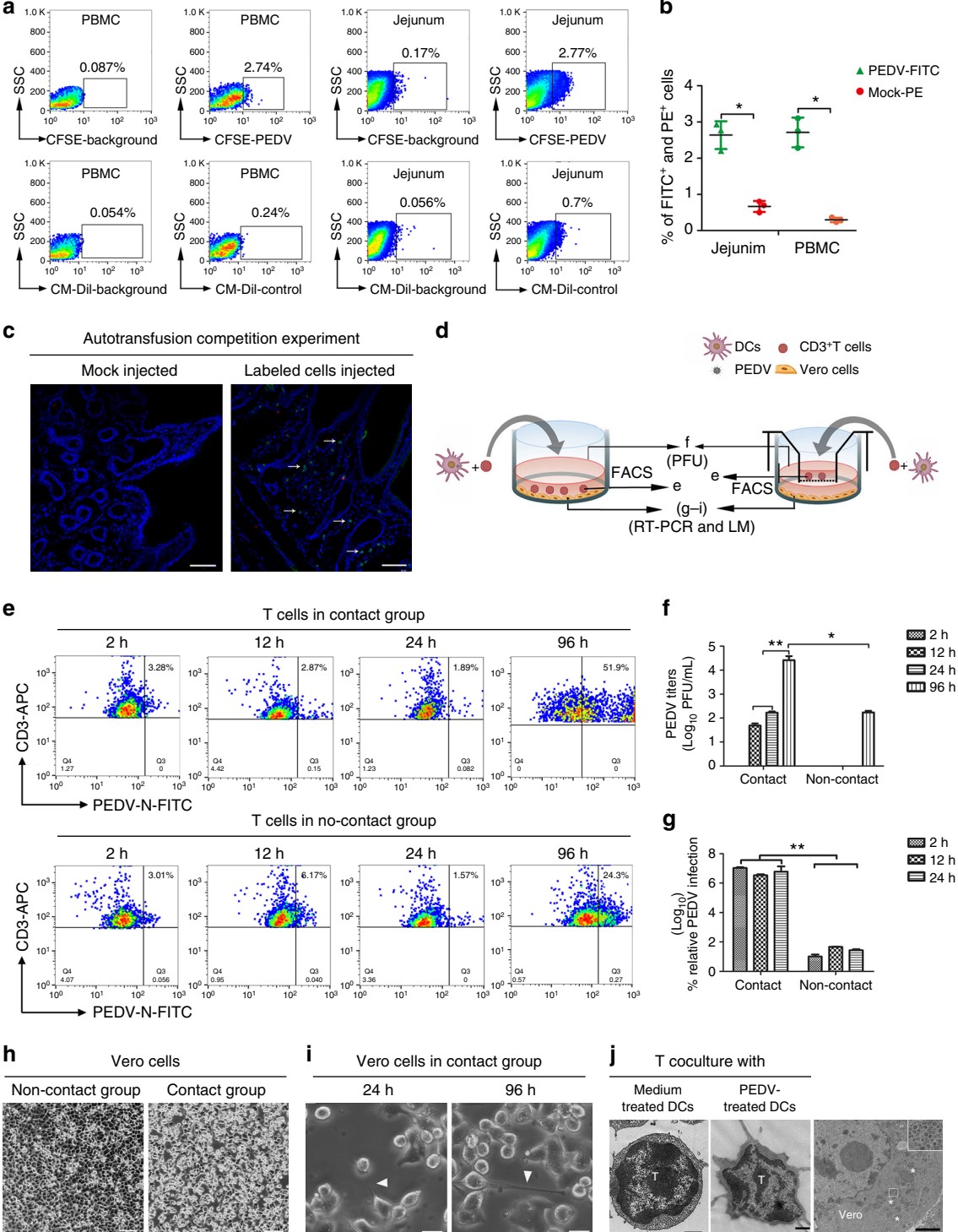

PEDV prevention and providing insights into the pathogenicity of viruses with the same characteristics, although the specific molecular mechanisms remain to be elucidated.

## Methods

**Virus**. The wild-type PEDV strain ZJ was obtained from intestinal contents of a 2-day-old diarrheic piglet on a farm in Jiangsu in 2012, and this strain clustered with the emerging virulent strain based on phylogenetic analysis. Viruses were purified using discontinuous sucrose density gradient centrifugation[55]. Heat-inactivated viruses were prepared at 56 °C for 0.5 h and tested for complete loss of infectivity by inoculation into Vero cells multiplicity of infection (MOI) = 1 for CPE observations. Fluorescence labeling of the virus was

performed with the fluorescent probe DyLight 633 NHS Ester (Thermo Fisher Scientific), according to the instructions provided by the manufacturer. Unincorporated dye was removed using commercial fluorescent dye removal columns (Thermo Fisher Scientific). Labeled viruses were stored at 4 °C and used within 2 days.

**Animals**. Caesarian-derived and colostrum-deprived xin huai piglets were obtained from a swine herd at the Jiangsu Academy of Agricultural Science (JAAS) and were artificially fed with milk. The swine herd was seronegative for antibodies against PEDV, PRRSV, PRCV, TGEV, and porcine circovirus type 2. Each experimental group of pigs was housed in a separate room in a high-security isolation facility. All animal procedures and experiments were performed according to protocols

**Fig. 7** PEDV-loaded T cells reach the intestine and mediate transfer infection. **a** Through an autotransfusion competition experiment, the percentage of PEDV-treated (CFSE) and PBS-treated (CM-DiI) cells among PBMCs and in the jejunum were analyzed by FACS. **b** The numbers of cells shown in (**a**) were quantified, $n = 3$ piglets per group. **c** After the autotransfusion competition experiment, frozen sections of the jejunum from mock-injected and fluorescence-labeled cells injected groups were stained with DAPI (blue) and observed by fluorescence microscopy. PEDV-treated (CFSE) CD3$^+$ T cells were marked with white arrowheads. Bars, 50 μm. **d** Schematic of the model to study PEDV-carrying CD3$^+$ T cell transmission of the virus to Vero cells. PEDV-pulsed DCs were cocultured with CD3$^+$ T cells for 2 h and donor DCs were removed by MACS. Then, the remaining CD3$^+$ T cells were cocultured with Vero cells by two methods (contact and noncontact coculture). **e** PEDV$^+$ CD3$^+$ T cells in the contact and noncontact groups were detected by FACS at the indicated times. **f** In addition, viral titers in the supernatant of the coculture system were measured by a plaque assay, $n = 3$. **g** PEDV RNA expression in Vero cells was evaluated by qRT-PCR at 96 h, $n = 3$. **h** At 48 hpi, Vero cells in the noncontact and contact groups were observed. LM light microscopy. **i** Vero cells (contact group) displayed thin cell bodies with long, branched membrane protrusions toward the T cells at 24 and 96 h. Bars, 20 μm. **j** TEM image showing the morphological features of sorted T cells cocultured with medium or PEDV-treated DCs. Bars, 1 μm. When cocultured for 96 h, Vero cells from the contact group were processed for TEM, which revealed an accumulation of virus particles in the cytoplasm. Some virus particles are indicated by a white asterisk. Bars, 1 μm. All data represent the mean ± SD, comparisons performed with $t$-tests (two groups) or analysis of variance (ANOVA) (multiple groups). *$P < 0.05$, **$P < 0.01$. Data were combined from at least three independent experiments unless otherwise stated

approved by the Institutional Animal Care and Use Committee of Nanjing Agricultural University (Nanjing, China) and followed the National Institutes of Health guidelines.

**Reagents and cell lines**. To study DCs maturation in coculture systems, fluorescently labeled anti-pig PE-SWC3a (1:200, ab25684), FITC-SWC3a (1:200, ab24885), PE-CD1a (1:200, ab25599) and the respective isotype and fluorochrome-matched control antibodies mouse IgG1 (SWC3a) and mouse IgG2a (CD1a) were purchased from Abcam. Anti-pig FITC-MHCII (1:400, MCA2314F) and its isotype control mouse and PE-IgG2b antibodies were purchased from Bio-Rad. For analyzing T cells, anti-pig APC-CD3ε (1:200, 561476) was purchased from BD Biosciences. The anti-PEDV N protein mAb was purchased from Medgene labs (FACS, 1:100; IF, 1:200; Western-blot, 1:1000). Porcine anti-PEDV polyclonal antibody was purchased from VMRD (1:200, PC-IFA-PEDV). The other antibodies included anti-mouse tight junction protein zonula occludens protein (ZO-1) mAb (1:200, Z01-1A12), which were cross-reactive with pig were purchased from Life Technologies. Anti-pig epithelial cell marker PE-Keratin 18 (CK18) mAb (1:200, NBP1-97715PE) were purchased from Novus Biologicals. For studying the NF-κB pathway, anti-pig p65 Rabbit mAb (1:1000, 93H1) and anti-pig p-P65 Rabbit mAb (1:1000, L8F6) were purchased from Cell Signaling Technology. Secondary antibodies used for IF, such as goat anti-mouse Alexa Fluor 488, goat anti-rat Alexa Fluor 594 and goat anti-rabbit Alexa Fluor 488, were purchased from Invitrogen. For cellular nucleus staining, Hoechst 33342 (1:2000, H3570) was purchased from Life Technologies. Anti-APC (130-097-143) and anti-PE MicroBeads (130-097-054) and MiniMACS Starting kits were all purchased from Miltenyi Biotec. Recombinant pig interleukin-4 (IL-4) protein (Z02928) and recombinant pig granulocyte-macrophage colony-stimulating factor (GM-CSF) Protein (711-pg-010) were from GenScript and R&D, respectively. CFSE (150347-59-4), CM-DiI Dye (C7001), and PKH26 Red Fluorescent Cell Linker Kit (PKH26GL) were from Sigma-Aldrich. All other reagents and chemicals, unless otherwise stated, were obtained from Sigma. Vero E6 cells (ATCC CCL81) were kindly provided by the Veterinary Medicine Research Center of the Da Bei Nong Group. The cell line was regularly tested for mycoplasma contamination.

**PEDV intranasal inoculation**. Female piglets (5 days old) with similar weight were allocated to 3 groups (3 piglets per group) with a completely random design using the random number generation function (Excel, Microsoft Corporation) and housed in three separate rooms 24 h prior to experiment (acclimation period). The three groups were (I) control, (II) PEDV intranasal inoculation, and (III) PEDV oral inoculation. Piglets in groups II and III were challenged with 1 ml PEDV ($10^7$ PFU ml$^{-1}$) by nasal and oral inoculation. The nasal spray device used for nasal inoculation is commonly used for vaccine absorption by the nasal mucosa and has good atomization effects. In group I, the same volume of PBS was inoculated as a negative control. The animals were artificially fed with milk every 3 h throughout the experiment to meet or exceed the National Research Council (NRC, 2012) requirements for nutrients and energy for this size pig. After a challenge, the piglets were observed daily for symptoms of diarrhea. The piglets were housed under the conditions of controlled humidity (50 ± 5%), temperature (30 ± 1 °C), and light (12-h light/12-h dark cycle, lights on at 8:00 AM). Investigators were not blinded to experimental groups. To further determine the dynamic changes in PEDV in piglets after nasal inoculation, we randomly assigned 18 piglets to two groups (I: control and II: PEDV intranasal inoculation), and the specific methods were the same as those mentioned above. Subsequently, three piglets from each group were euthanized at 3, 12, and 24 h after virus inoculation. Investigators were not blinded to experimental groups.

**Generation of T cells and DCs**. Porcine PBMCs were isolated from the blood of piglets by density centrifugation using a porcine peripheral blood lymphocyte

separation kit (Solarbio). The morphology of DCs is shown in Supplementary Fig. 3a. To isolate T cells, PBMCs were labeled with APC-CD3 antibody, incubated with anti-APC microbeads and sorted using MiniMACS Starting kits. Swine DCs were obtained from the bone marrow of piglet femurs according to methods that had been optimized by our laboratory[19]: bone marrow was obtained from femurs of 2-week-old piglets by puncture method and treated with red blood cell lysing buffer. The bone marrow cells were differentiated into DCs by resuspending the cells in complete medium Roswell Park Memorial Institute (RPMI) 1640 medium supplemented with 10% FBS, 1% penicillin/streptomycin, 20 ng ml$^{-1}$ porcine GM-CSF, and 10 ng ml$^{-1}$ porcine IL-4 and plated at $10^6$ cells ml$^{-1}$ in 6-well plates. Non-adherent granulocytes were removed by discarding the culture medium after 60 h. On 6 days culture, the clusters were harvested and subcultured overnight for removing adherent cells. Non-adherent cells were collected after 7 days, washed, and used as DCs for the subsequent studies.

**Establishment of the NECM and NECM/DCs coculture system**. The NECs culture protocol was based on a modified method used for human airway epithelial cultures[56]. The acquired NECs were collected, adjusted to $10^6$ cells ml$^{-1}$ and seeded at a concentration of $2 \times 10^5$ cells cm$^{-2}$ onto Transwell tissue culture inserts (3 μm pore size; 6.5 mm membrane diameter; Corning) coated with collagen from human placenta Type IV (6 μg ml$^{-1}$) (Sigma). After the cells reached confluence, the medium was replaced with a differentiating medium, and the cells were brought to the air–liquid interface. When the NECM formed an air–liquid interface and stable TEER, the filters were turned upside down, and the DCs ($5 \times 10^5$) were cultured for 4 h on the filter facing the basolateral membrane of NECs to allow the cells to attach to the filter. Medium, DyLight 633-labeled PEDV, and WIV PEDV were incubated on the apical side of the NECs and cultured at 37 °C for 1 h. Then, DCs were collected for FACS analyses. Moreover, DyLight 633-labeled PEDV were incubated on the apical side of the NECs at MOI of 0.1 and 0.5, and the coculture system was collected for CLSM analyses. As a control group, DyLight 633-labeled PEDV were incubated at 4 °C with a MOI of 0.1.

**PEDV infection and transmission**. DCs were pulsed with PEDV (MOI = 0.1) for 1 h at 37 °C and washed extensively to remove unbound virus. PEDV-pulsed DCs were cocultured with autologous positively selected CD3$^+$ T cells that had been maintained at 37 °C in RPMI 1640 medium supplemented with IL-2 (10 IU ml$^{-1}$) for the subsequent study. After 2 h, donor DCs were removed from CD3$^+$ T cells using MiniMACS Starting kits (anti-Swc3a-PE and anti-PE beads). PEDV-carrying CD3$^+$ T cells were cultured in maintenance medium for viral RNA titer determination. For TEM studies, the mixtures were centrifuged at 200×g for 1 min to facilitate conjugate formation prior to co-culturing. After 1 h in culture, the mixtures were collected by centrifugation at 300×g for 5 min and fixed in glutaraldehyde/cacodylate buffer.

**Conjugate formation**. The DCs obtained after 7 days of culture, untreated or treated with PEDV (MOI 0.1), were labeled with PKH26 according to the manufacturer's instructions. T cells were labeled with 10 nM CFSE for 10 min at 37 °C. The PEDV-pulsed DCs and T cells mixtures were centrifuged at 200×g for 5 min. The pellet was incubated at 37 °C for 30 min, gently resuspended in 200 μl of PBS and analyzed immediately for double-labeled cell conjugates by flow cytometry (BD FACS Calibur).

**Transfer infection**. Vero cells were seeded in 24-well plates and grown to monolayers before transfer infection. PEDV-carrying CD3$^+$ T cells were cocultured with Vero cells by two methods (contact and noncontact). In the contact method, T cells were directly cultured with Vero cells, whereas in the noncontact method, T cells were grown in Transwell™ assemblies with a pore diameter of 0.4 μm over Vero cell monolayers. Negative controls, including uninfected T cells, and

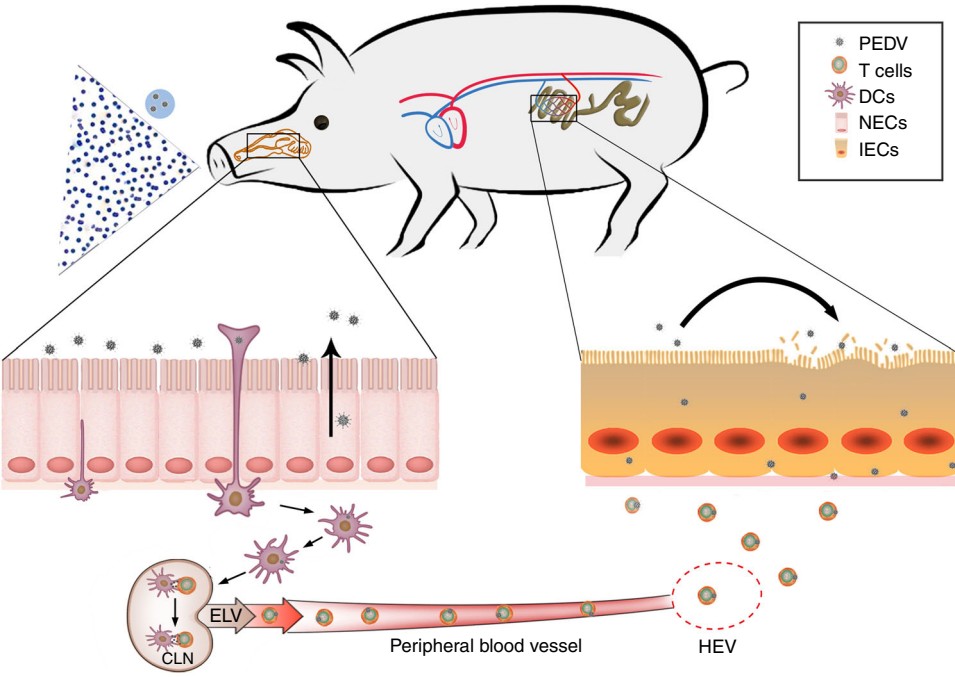

**Fig. 8** Schematic of the proposed mechanism for PEDV transportation in swine. When airborne PEDV enters the nasal cavity, the virus accumulates and propagates in NECs and is released from the apical side of NECs. DCs may play an important role in helping the virus enter the nasal mucous and be transferred to CD3+ T cells. Furthermore, virus-carrying T cells enter the blood and reach the intestine through lymphocyte recirculation. Finally, the virus-carrying CD3+ T cells can transfer the virus to intestinal epithelial cells, causing typical PED symptoms. ELV efferent lymphatic vessels, CLN Cervical lymph node, IECs intestinal epithelial cells, NECs nasal epithelial cells, HEV high endothelial venules

positive controls, which consisted of an identical number of PEDV viruses placed over the 0.4 μm filters, were used to verify that infectious cell-free PEDV was able to cross the filters in the Transwell™ assemblies. T cells and the medium of the Vero coculture were collected at the indicated times. Vero cells were washed five times with PBS to eliminate the nonabsorbed virus and T cells, followed by culture in maintenance medium for 96 h.

**In vivo competition experiments**. For the in vivo competition assay, CD3+ T cells were isolated from three littermate piglets and cocultured with DCs (treated or not with PEDV). Then, the CD3+ T cells were purified and labeled with CFSE (green) or CM-DiI (red, Molecular Probes), respectively, for long-term cell labeling. Equal numbers of cells ($1 \times 10^7$) from the two populations were then mixed and adoptively transferred into piglets (autologous) via front cavity vein injection. The recipient piglets were sacrificed 24 h after the injection. For the flow cytometry analyses, cells were isolated from PBMCs, duodenum, jejunum, and ileum. For histological analyses, the duodenum, jejunum, and ileum were washed with 0.1 M PBS (pH 7.4) and embedded in OCT compound (Sakura Finetechnical). The frozen tissues were cut into 10-μm sections and mounted onto poly-L-lysine-coated glass slides. The CM-DiI or CFSE-labeled cells were observed with a fluorescence microscope (Carl Zeiss). Investigators were not blinded to experimental groups. Pathologists interpreting slides were blinded to PEDV-treated and control animal identifiers.

**Quantitative RT-PCR**. Total RNA from different tissues and Vero cells was extracted using TRIzol reagent (Invitrogen) according to the manufacturer's instructions. cDNA was generated by reverse transcription using HiScript TM QRT SuperMix (Vazyme) according to the manufacturer's instructions. qRT-PCR was performed with a SYBR Green qPCR Kit (TaKaRa) in the Applied Biosystems 7500 Fast Real-Time PCR System (Life Technologies). The gene expression levels were normalized to those of glyceraldehyde 3-phosphate dehydrogenase (GAPDH). PEDV load was determined by detecting the viral membrane (M) gene and analyzed by the double standard curve method. PCR products were cloned into the pJET1.2 vector (Thermo Fisher Scientific). Plasmids were serially diluted and used as standards for the quantitative analysis. The initial copy number of the PEDV M gene and GAPDH in each group was calculated using the following formula: $X0 = -K \log Ct + b$, where $X0$ is the initial copy number; and K, Ct, and b refer to the slope rate, cycle threshold, and constant, respectively. All of the amplification primers used are listed in Supplementary Table 1.

**Viral titer and Western blot analysis**. The viral titer of supernatant samples was measured by plaque assays. Confluent monolayers of Vero cells grown in 6-well tissue culture plates were infected with 500 μl of serial ten-fold dilutions of the

supernatant samples. After incubation for 1 h at 37 °C, cells were overlaid with 0.7% agarose in Dulbecco's Modified Eagle's Medium (DMEM) containing 2% fetal bovine serum (FBS) and incubated at 37 °C. At 3 days postinfection, plaques were visualized by staining with Crystal Violet.

Cell samples were assessed by Western blotting with specific antibodies. The expression of GAPDH was detected with an anti-GAPDH mouse mAb to demonstrate equal protein sample loading. Uncropped Western blot images of data shown in Figs. 1 and 5 can be found in Supplementary Figures 4.

**Flow cytometric analysis**. In vivo, after PEDV inoculation, epithelial cells were acquired from the mucosa of the nasal cavity and stained with an antibody against CK18. After surface staining, the cells were resuspended in fixation/permeabilization solution (BD Cytofix/Cytoperm kit, BD Pharmingen) and stained with PEDV N protein antibody to detect intracellular PEDV. PBMCs were similarly stained with an antibody specific for porcine CD3 and PEDV N protein. The collected DCs were detected by FACS for fluorescent PEDV uptake. In addition, after NECM/DCs coculture, the cells were stimulated with PEDV for 24 h. The harvested DCs were washed twice with cold PBS and stained with fluorescent mAbs specific for porcine MHCII, SWC3a and CD1a or the respective isotype controls at 4 °C for 0.5 h, according to the manufacturer's guidelines. After three washes with PBS, the cells were phenotypically analyzed by FACS (BD FACS Calibur).

**Cytokine assays by enzyme-linked immunosorbent assay (ELISA)**. Cytokines were analyzed by ELISA. The production of cytokines (CCL25, CCL20, and CXCL2) was measured using ELISA kit (eBioscience), which was performed according to the manufacturer's instructions.

**Immunohistochemistry and IFA assay**. PEDV-infected piglets were anesthetized and sacrificed via an intravenous injection of pentobarbital sodium (100 mg kg$^{-1}$) 12 h post intranasal inoculation. After fixation, histological sections of four blocks were selected according to the fractions 1/4, 2/5, 3/5, and 4/5, using the landmarks provided in the diagram presented in Supplementary Fig. 1a. Four cross-sections (I, II, III, and IV) taken at the fractions (Supplementary Fig. 1b) were subsequently selected for assessing the distribution of PEDV by immunohistochemistry (IHC) using primary antibodies directed against PEDV N protein. DCs located in nasal mucosa were immunolabeled with rat anti-Swc3a mAb and mouse anti-MHCII mAb, followed by Alexa Fluor 594-conjugated goat anti-rat IgG and Alexa Fluor 488-conjugated goat anti-mouse IgG. The colonization of PEDV in cross-sectional sections IV was detected by IFA with PEDV polyclonal antibody. Fixed filters were permeabilized in 0.2% Triton X-100 in PBS for 5 min. After blocking with 5% bovine serum albumin in PBS for 1 h, the filters were incubated with primary antibodies overnight at 4 °C,

followed by fluorescent secondary antibodies at room temperature for 1 h. Pathologists interpreting slides were blinded to PEDV-treated and control animal identifiers.

**Statistical analysis**. Results are expressed as the means ± SD and analyzed with SPSS 17.0. One-way analysis of variance (ANOVA) was employed to determine significant differences among multiple groups, and a *t*-test was employed to determine the differences between the two groups. *$P < 0.05$, **$P < 0.01$. Data were combined from at least three independent experiments unless otherwise stated.

## Data availability

The data that support the findings of this study are available from the corresponding author upon request.

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

## Acknowledgements

We thank Veterinary Medicine Research Center of the Da Bei Nong Company and Dr. Zhixing Feng (Jiangsu Academy of Agricultural Sciences, JAAS) for the help in our experiments; Professor Qinghua Yu and Professor Jian Lin for helpful discussion and suggestions; and Dr. lvfeng Yuan for technical support. This work was supported by the National Science Grant of China (31772777) and a project funded by the Priority Academic Program Development of Jiangsu Higher Education Institutions (PAPD).

## Author contributions

Yuchen Li was responsible for performing the experiments, data analysis, and writing the manuscript. Qinxing Wu and Jialu Wang were responsible for animal experiments. Lulu Huang and Chen Yuan were responsible for a series of coculture models establishment. Qian Yang was responsible for the conception and design of the study, data collection, drafting the article, and final approval of the version submitted.

## Additional information

**Competing interests:** The authors declare no competing interests.

