## [Peer Review File · Nature Communications]

Reviewers' comments:

Reviewer #1 (Remarks to the Author):

The manuscript "A novel pathway of virus dissemination within the host: spread of porcine epidemic diarrhea virus (PEDV) from the nasal cavity to the intestinal mucosa in swine" describes an alternative possible pathogenesis route by which Porcine Epidemic Diarrhea virus (PEDV) can result in active infection. The authors demonstrate that the intranasal infection of pigs results in typical clinical signs of PEDV, and provide insight on the possible role of dendritic cells in the nasal cavity to transfer virus to T lymphocytes that then possibly work as carriers to the active site of infection (intestines). The aerosol transmission of PEDV has been previously speculated as an important factor to disseminate the virus. However, although PEDV was found in aerosol particles, transmission and active replication was not demonstrated in previous studies. This manuscript provides novel evidence to the mechanism in which PEDV could reach the active site of replication if infection was not initiated by the oral route. The methodology was thorough and well performed, but clarity needs to be improved for parts of the methods and most of the results/figures. Specific comments are mentioned below.

Porcine jejunum epithelial cell line (IPEC-J2) is mentioned for the transfer experiment methods, but results are not shown. These results are crucial to confirm the ability of T lymphocytes to transfer PEDV to intestinal cells, as they are a host-specific cell line and provide results that can be better extrapolated to what happens in vivo than Vero cells.

Also, for the NEC-DC and DC-T cell co-culture studies, were there controls to compare the direct infection of DC and T lymphocytes? Does it result in higher percentage of infected cells if they are directly infected?

The authors mention that large amounts of virus were transferred from infected Vero cells to T cells. Is this a hypothesis for the higher virus titers in the supernatant of the transfer experiment at 96hrs? Please explain.

Overall the manuscript is well written, but I believe it would be benefited if edited for proper grammar and spelling. Some examples are listed below:

L98: Replace "are possessing" by possess.

L133: "Different parts of the body" should be replaced by different organs or tissues.

L143: Remove "subsequently".

L146-147: were detected in the jejunum and ileum at higher levels than in other tissues.

L149: nasal mucosa of piglets were isolated at different time points after intranasal infection.

L158-160: in which most DCs and other lymphocytes accumulated

L189: Remove while. The word was used unnecessarily in additional sentences in the manuscript.

L216: After 12h post administration of PEDV. By using the expression "for 12h" seems that the infection lasted 12h.

The methodology and presentation of results are often confusing, mostly due to the excessive use of abbreviations, poorly described figures, or lack of descriptive details. The figures and results contradict the methods section at times. This could be an obstacle for reproducibility of the work. Some examples are given below:

L106-107: Following sacrifice for macroscopic examination at PIDs 3. Was this at 3 days post infection? If so, there was no mention that animals were euthanized at 3 days post-infection.

Overall, the initial experiment for clinical signs (oral and nasal inoculation) was not well described. The methods for the co-culture of NEC and DC are not complete of how NEC were collected and the procedure for infection, most of the information is only mentioned on the figure legend but is not necessarily clear.

Abbreviations are used extensively and loosely. Some were not described at all (e.g. PID, TEER in the supplementary figure only). And even those that were mentioned previously, if not commonly known or extensively repeated in the manuscript, the meaning should be reinforced. Some are mentioned in the methods section but will need to be mentioned in the results too.

Because of the way the manuscript is structured (results before the methods), I believe it would be more clear and easier to follow results if before displaying the findings, a brief explanation was given of why a particular assay was done.

The targets and reason of antibodies/ markers used should be mentioned when describing antibodies, assuming the reader will not be familiar with all of them.

The 1h time point shown for the PEDV infection/transmission is confusing. The methods describe co-culture for 2h. Please describe timepoints in figure legends more clearly.

Figures:

Legends are not clear, it is necessary to re-check methods and text frequently to understand figures. What is shown in the figures should be more clearly explained. In several instances, the panels are too small and difficult to visualize, most frequently for fluorescent staining.

Fig 3: Position of letters indicating panels are not correct within a sentence, which makes it confusing. Replace electron microscopy (G) by electron microscopy (E).

Fig S3: the order of letters for panels are not correct, and not all are mentioned.

Fig. 4: What is the PEDV 4°C?

Fig. 6: the time points shown in A and B panels are different. Co-culture was for 2hs, not for 1h as mentioned. Asterisks are black, not white and there are only arrowheads, not arrows.

Fig. 7: Panels C and D descriptions are switched. There is no (e) panel.

General comments:

Substitute "gene expression" by RNA expression or RNA levels/titers throughout.

L154: Remove gradually. There was no gradual decline observed since only one more time point was measured. Best if say: level of virus in NECs declined at 60h. Also, it is better if time points after infection are referred as hours post infection.

L177: Replace in vivo by in vitro.

L263-264: Panel letters are switched, J is shown in L, K is shown in M.

L313-314: PEDV and DCs are switched.

L434: Replace score with titers.

L435: The longer incubation period observed in the group inoculated intranasally

L438: Nasal mucosa

L448-450: Move the last sentence to the next paragraph.

The sentence "as previously described" is used several times without referencing.

Reviewer #2 (Remarks to the Author):

This manuscript describes the detail of a novel pathogenic pathway for porcine epidemic diarrhea virus infection in pigs.

While an aerosol transmission of virus has been hypothesised previously, this manuscript offers evidence to support the hypothesis, in the form of in vivo and in vitro infections and dissemination studies.

The level of detail in this investigation opens new avenues for research in this area, as well as stimulating further consideration of PED transmission in the field.

The manuscript is well structured and the figures are well presented to support the results and discussion.

Some additional detail in the methods section would be useful:

1) Lines 559-561: Please indicate how much virus was used here.

2) Lines 568-570: What sex animals were used?

3) Lines 598-600 and 604-608: Please specify how the randomisation was carried out.

4) Lines 600-602: Please indicate details of the nasal spray device used.

5) Lines 603-604: Were any welfare checks carried out on the animals to minimise unnecessary suffering?

6) Lines 651-653 and 718-719: Please indicate the instrument used for flow cytometry analysis.

7) Lines 673-675: Please indicate the site of the IV injection.

- 8) Line 702: Please specify the type of cells was used for the plaque assay.
- 9) Lines 728-731: Please specify which tissues were collected for this analysis.

As a very minor point, the reference format used in lines 53-54 is inconsistent with the rest of the manuscript.

Manuscript NCOMMS-18-07999

Title: A novel pathway of virus dissemination within the host: spread of porcine epidemic diarrhea virus (PEDV) from the nasal cavity to the intestinal mucosa in swine.

Referee #1 (Remarks to the Author):

The manuscript “A novel pathway of virus dissemination within the host: spread of porcine epidemic diarrhea virus (PEDV) from the nasal cavity to the intestinal mucosa in swine” describes an alternative possible pathogenesis route by which Porcine Epidemic Diarrhea virus (PEDV) can result in active infection. The authors demonstrate that the intranasal infection of pigs results in typical clinical signs of PEDV, and provide insight on the possible role of dendritic cells in the nasal cavity to transfer virus to T lymphocytes that then possibly work as carriers to the active site of infection (intestines). The aerosol transmission of PEDV has been previously speculated as an important factor to disseminate the virus. However, although PEDV was found in aerosol particles, transmission and active replication was not demonstrated in previous studies. This manuscript provides novel evidence to the mechanism in which PEDV could reach the active site of replication if infection was not initiated by the oral route. The methodology was thorough and well performed, but

clarity needs to be improved for parts of the methods and most of the results/figures. Specific comments are mentioned below.

Thank you very much for your approval and opinions. These suggestions will be helpful in improving the quality of our manuscript. We will answer your specific questions individually, perform the relevant experiments and correct a series of errors in the manuscript. All the changes we made are labeled with red font in the revised manuscript.

Q1. Porcine jejunum epithelial cell line (IPEC-J2) is mentioned for the transfer experiment methods, but results are not shown. These results are crucial to confirm the ability of T lymphocytes to transfer PEDV to intestinal cells, as they are a host-specific cell line and provide results that can be better extrapolated to what happens *in vivo* than Vero cells.

A: I apologize for the missing results. Actually, we used the porcine jejunum epithelial cell line (IPEC-J2) for the transfer infection experiment. No viral protein and infectious virus particle was detected in the IPEC-J2 in both the noncontact and direct contact groups throughout the study (Fig. R1). In additional studies, IPEC-J2 did not appear to be susceptible to PEDV, consistent with the previous reports¹. After PEDV infection, no infectious virus particles were released in the cell culture supernatant (Fig. R2A), and the viral RNA and protein in IPEC-J2 cells gradually disappeared over time (Fig. R2B and C). We did not think that IPEC-J2

could be used as an appropriate cell model for transfer infection experiments. Therefore, these results were not included in the article. We will also delete the corresponding content in the Materials and Methods section of the manuscript.

Compared with IPEC-J2, Vero cells support the serial propagation of PEDV and grow successfully *in vitro*; these cells are commonly used for PEDV isolation, propagation and pathogenesis research^{2,3,4}. Vero cells can also be a suitable model for studying PEDV transfer infection in the absence of an adequate host-specific cell line. Thanks very much for your careful review.

Fig. R1. PEDV-loaded CD3⁺T cells can not mediate transfer infection in IPEC-J2 cells. After coculture with PEDV-carrying CD3⁺T cells at the indicated time, IPEC-J2 were digested with trypsin and analyzed by FACS. No PEDV⁺IPEC-J2 cells was detected in contact (A) and noncontact (B) groups. (C) Meanwhile, virus titers in the supernatant of the coculture system were detected by plaque assay. Compared with CD3⁺ T/Vero coculture system, no virus was detected in the supernatant of CD3⁺/IPEC-J2 coculture system.

Fig. R2. Infectivity of PEDV in Vero and IPEC-J2 cells. (A) The viral titers in the supernatant of PEDV infected IPEC-J2 and Vero cells were assayed by a plaque assay. (B) Immunofluorescence images of PEDV (green) infected IPEC-J2 and Vero cells at indicated time. Cell nuclei was stained with DAPI (blue). (C) RNA expression of PEDV in IPEC-J2 and Vero cells (MOI = 0.1) at different times. The scale bars represent 20 μm . Data express means \pm SD (n = 3).

Q2. Also, for the NEC-DC and DC-T cell co-culture studies, were there controls to compare the direct infection of DC and T lymphocytes? Does it result in higher percentage of infected cells if they are directly infected?

A: Thanks very much for your helpful suggestion. We have performed relevant experiments to study the direct infection of DCs. A supplementary experiment showed that compared with the NECM/DCs coculture system, PEDV direct infection resulted in a higher percentage of PEDV⁺DCs (Fig. R3A). This result could be due to the epithelial barrier in the NECM/DCs coculture system in which DCs used tight junction proteins to penetrate the epithelium and capture PEDV. Our results are consistent with previous report⁵.

Additionally, we conducted experiments to study PEDV infection in T cells. A higher percentage of PEDV⁺ CD3⁺T cells (Fig. R3B) was detected in the direct infection group compared with that in the DC/T coculture system (Fig. 6C). If you find it necessary, we will add this result to the manuscript; we would appreciate your suggestions.

Fig. 3R. (A) After direct infection, the PEDV⁺ DCs cells at 1 h post infection were detected by FACS. DCs were gated based on SWC3a⁺ cells, and viral infection was detected by PEDV N protein staining. (B) The percentage of PEDV infected CD3⁺T cells at 2 h post direct infection was detected by FACS. T cells were gated based on CD3⁺ cells, and PEDV infection was detected by PEDV N protein staining.

Q3: The authors mention that large amounts of virus were transferred from infected Vero cells to T cells. Is this a hypothesis for the higher virus titers in the supernatant of the transfer experiment at 96hrs? Please explain.

A: I apologize for the confusing statement. Because the coculture system contained both T and Vero cells, high virus titers in the supernatant was not enough to conclude that PEDV could be transferred to T cells. We speculated that large amounts of virus were transferred from infected Vero cells to T cells based on the flow cytometry analysis results in Fig. 7F. In the T/Vero coculture system, 3% PEDV⁺CD3⁺ T cells were detected in both the contact and noncontact groups at the original stage. A larger number of PEDV-positive T cells appeared in both groups after 96 h of coculture, particularly in the contact group, and as much as 51.9% of PEDV⁺ CD3⁺ T cells were detected. However, our data are insufficient to confirm the conclusion about large amounts of virus were transferred from infected Vero cells to T cells. We need apologize for our hasty conclusion. We have conducted a complementary experiment to prove that PEDV could be transferred from infected Vero cells to T cells (Fig. R4). If you find it necessary, we will include the relevant information in the revised manuscript as supplementary material. We appreciate your useful suggestion. Thanks.

Fig. R4. PEDV can be transferred from infected Vero cells to CD3⁺T cells. (A) Schematic of the model to study transmission of PEDV from Vero cells to CD3⁺T cells. 70% confluent Vero cells were inoculated with PEDV at a MOI of 0.1 for 1 h at 37°C. The unattached virus were removed and fresh growth medium was added. Infected Vero cells were continued to incubate at 37°C for 6 h and washed with PBS (pH7.2 at 4 °C) three times to remove unattached virus. Then CD3⁺ T cells were cocultured with Vero cells (direct contact) for indicated time. (B) PEDV in CD3⁺ T cells was detected by FACS at 2 h, 18 h, 30 h and 42 h. (C) After culturing for 42 h, the coculture system were stained with anti-CD3 and PEDV N antibodies, and examined by IF microscopy. PEDV-carried CD3⁺T cells were marked with white arrowheads. Red, CD3⁺T; green,

PEDV. (D) After coculturing with Vero cells for 30 h, CD3⁺T cells were separated by magnetic beads separation (MACS) and culturing for indicated time. PEDV RNA expression levels were detected by Real-time PCR, the results of a representative experiment (n=3) are shown.

Q4: Overall the manuscript is well written, but I believe it would be benefited if edited for proper grammar and spelling. Some examples are listed below:

L98: Replace “are possessing” by possess.

L133: “Different parts of the body” should be replaced by different organs or tissues.

L143: Remove “subsequently”.

L146-147: were detected in the jejunum and ileum at higher levels than in other tissues.

L149: nasal mucosa of piglets were isolated at different time points after intranasal infection.

L158-160: in which most DCs and other lymphocytes accumulated.

L189: Remove while. The word was used unnecessarily in additional sentences in the manuscript.

L216: After 12h post administration of PEDV. By using the expression “for 12h” seems that the infection lasted 12h.

A4: Thanks very much for your helpful suggestions and careful review. We have modified all the mistakes in our manuscript according to your suggestions. Moreover, a native English language editing institute has been commissioned to edit our article. The grammar, spelling, and punctuation have been verified and corrected where needed and are labeled in yellow font in the revised manuscript. The certificate of English language editing has been attached to the supplementary materials.

Q5: The methodology and presentation of results are often confusing, mostly due to the excessive use of abbreviations, poorly described figures, or lack of descriptive details. The figures and results contradict the methods section at times. This could be an obstacle for reproducibility of the work. Some examples are given below:

A5: Thanks for your helpful suggestion. We have carefully checked the manuscript, including the methods, figure legends and results. We have corrected the contradicting sections and added more details in the revised manuscript. Thanks!

1) L106-107: Following sacrifice for macroscopic examination at PIDs 3.

Was this at 3 days post infection? If so, there was no mention that animals were euthanized at 3 days post-infection. Overall, the initial

experiment for clinical signs (oral and nasal inoculation) was not well described.

A: I apologize for the missing details. We have added the detailed information regarding piglet sacrifice in the “Materials and Methods” section. Moreover, the clinical signs of piglets after PEDV inoculation (oral and nasal) were rewritten in the revised manuscript to provide a more detailed description.

2) The methods for the co-culture of NEC and DC are not complete of how NEC were collected and the procedure for infection, most of the information is only mentioned on the figure legend but is not necessarily clear.

A: Thanks for your suggestion. I apologize for the missing details concerning NECs collection and the procedure for PEDV infection. We have added specific information about these procedures in the “Materials and Methods” section of the revised manuscript.

Q6: Abbreviations are used extensively and loosely. Some were not described at all (e.g. PID, TEER in the supplementary figure only). And even those that were mentioned previously, if not commonly known or extensively repeated in the manuscript, the meaning should be reinforced. Some are mentioned in the methods section but will need to be mentioned in the results too. Because of the way the manuscript is

structured (results before the methods), I believe it would be more clear and easier to follow results if before displaying the findings, a brief explanation was given of why a particular assay was done.

A6: Thanks for your helpful suggestions. We have modified all of the abbreviations used in the revised manuscript. Moreover, we have added a brief explanation about a particular assay before displaying the findings.

Q7: The targets and reason of antibodies/ markers used should be mentioned when describing antibodies, assuming the reader will not be familiar with all of them.

A7: Thanks for your suggestion. We have added the targets and reasons for using antibodies/ markers in the “Materials and Methods” section.

Q8: The 1h time point shown for the PEDV infection/transmission is confusing. The methods describe co-culture for 2h. Please describe timepoints in figure legends more clearly.

A8: I apologize for our carelessness. For analysis of transmission of DC-associated PEDV to T cells, PEDV transmission was quantified after coculture for 2 h. We have added the timepoint to the figure legends of Fig. 6C and corrected the time point in related figures. However, for better observation of conjugate formation by transmission electron

microscopy (TEM), PEDV–pulsed DC–T-cell mixtures were centrifuged at $500 \times g$ for 1 min to facilitate conjugate formation and cultured for 1 h at 37°C as previously described⁶. We have also corrected the time point in the legend of Fig. 6H and the related section in the Materials and Methods.

Q9: Figures: Legends are not clear, it is necessary to re-check methods and text frequently to understand figures. What is shown in the figures should be more clearly explained. In several instances, the panels are too small and difficult to visualize, most frequently for fluorescent staining.

A: I apologize for the unclear figure description and presentation. We have checked all of the figure legends and corrected the unclear presentation. Furthermore, we enlarged small figures such as Fig 4. B and F, Fig. 5H, Fig. 6G, Fig. 7C and D for better visualization. Thanks again for your helpful suggestion.

1) Fig 3: Position of letters indicating panels are not correct within a sentence, which makes it confusing. Replace electron microscopy (G) by electron microscopy (E).

A: I apologize for our mistake. We have replaced electron microscopy (G) with electron microscopy (E).

2) Fig S3: the order of letters for panels are not correct, and not all are mentioned.

A: Thanks for your suggestion. We have corrected the order of the letters in Fig S3.

3) Fig. 4: What is the PEDV 4°C?

A: Thanks for your careful review. PEDV 4°C means that the NECM/DC coculture system was cultured at 4°C after PEDV infection. We have added the specific description to the legend of Figure 4E.

4) Fig. 6: the time points shown in A and B panels are different. Co-culture was for 2hs, not for 1h as mentioned. Asterisks are black, not white and there are only arrowheads, not arrows.

A: I apologize for our mistakes. We have corrected all of the mistakes you mentioned. Thanks again for your helpful suggestion.

5) Fig. 7: Panels C and D descriptions are switched. There is no (e) panel.

A: Thanks for your suggestion. We have corrected the order of the panels.

General comments:

1) Substitute “gene expression” by RNA expression or RNA levels/titers throughout.

A: Thanks for your suggestion. We have replaced “gene expression” with RNA expression throughout the manuscript.

2) L154: Remove gradually. There was no gradual decline observed since only one more time point was measured. Best if say: level of virus in NECs declined at 60h. Also, it is better if time points after infection are referred as hours post infection.

A: We agree with your suggestion. We have removed gradually and replaced time points after infection with hours post infection. Thanks.

3) L177: Replace in vivo by in vitro.

A: I am sorry for the mistake. We have replaced *in vivo* with *in vitro* in this sentence.

4) L263-264: Panel letters are switched, J is shown in L, K is shown in M.

A: Thanks. We have corrected the panel letters.

5) L313-314: PEDV and DCs are switched.

A: Thanks for your suggestion. We have corrected this mistake.

6) L434: Replace score with titers.

A: Thanks for your suggestion. We have replaced score with titers.

7) L435: The longer incubation period observed in the group inoculated intranasally

A: Thanks. We have corrected this sentence.

8) L438: Nasal mucosa

A: I am sorry for this mistake. We have corrected it .

9) L448-450: Move the last sentence to the next paragraph.

A: We completely agree with your opinion. We have moved the last sentence to the next paragraph.

10) The sentence “as previously described” is used several times without referencing.

A: I apologize for the missing references. We have added the relevant references. Thanks.

1. Reviewer #2 (Remarks to the Author):

This manuscript describes the detail of a novel pathogenic pathway for porcine epidemic diarrhea virus infection in pigs. While an aerosol transmission of virus has been hypothesised previously, this manuscript offers evidence to support the hypothesis, in the form of in vivo and in vitro infections and dissemination studies. The level of detail in this investigation opens new avenues for research in this area, as well as stimulating further consideration of PED transmission in the field. The manuscript is well structured and the figures are well presented to support the results and discussion. Some additional detail in the methods section would be useful:

Thank you very much for your approval and careful review, which will be beneficial to improve the quality of our manuscript. Our replies are shown below, including point-by-point answers to all your questions, the addition of the missing details and correction of a series

of errors. All the changes we made are labeled with red font in the revised manuscript.

1) Lines 559-561: Please indicate how much virus was used here.

A: I am sorry for the missing details. We inoculate PEDV at a MOI of 1. We have added the missing information in this sentence.

2) Lines 568-570: What sex animals were used?

A: Thanks for your suggestion. We used female piglets for the animal experiment. We have added the missing information in this sentence.

3) Lines 598-600 and 604-608: Please specify how the randomisation was carried out.

A: Thanks for your helpful suggestions. We chose and grouped the animals using the random number generation function of EXCEL. We have added the missing information in both sentences.

4) Lines 600-602: Please indicate details of the nasal spray device used.

A: Thanks for your suggestion. The nasal spray device that we used has good atomization effects and is commonly used for vaccine absorption through the nasal mucosa. We have added the

missing information in the revised manuscript. The device is also shown in the figure below.

5) Lines 603-604: Were any welfare checks carried out on the animals to minimise unnecessary suffering?

A: Thanks for your helpful suggestion. Our animal studies were approved by the appropriate Institutional Animal Care and Use Committee of Nanjing Agricultural University (Nanjing, China) and followed the National Institutes of Health guidelines. We performed various measures on the animals to minimize unnecessary suffering including but not limited to artificially feeding with milk to meet the requirements for nutrients and controlled humidity, temperature and light to ensure piglet

comfort. We have added the specific information in the revised manuscript.

- 6) Lines 651-653 and 718-719: Please indicate the instrument used for flow cytometry analysis.

A: Thanks for your suggestion. We used BD FACS Calibur for flow cytometry analysis. We have added the instrument information in this section.

- 7) Lines 673-675: Please indicate the site of the IV injection.

A: Thanks for your suggestion. The site we chose for IV injection was the front cavity vein, and we have added the missing information in this section.

- 8) Line 702: Please specify the type of cells was used for the plaque assay.

A: I apologize for the missing information. We used Vero cells for the plaque assay. We have added the cell type in this section.

- 9) Lines 728-731: Please specify which tissues were collected for this analysis.

A: Thanks for your suggestion. We have added the specific tissues collected for IHC analysis in this section.

- 10) As a very minor point, the reference format used in lines 53-54 is inconsistent with the rest of the manuscript.

A: I apologize for the inconsistently formatted references. We corrected the format of this reference. Thanks.

References:

1. Zhang Q, Yoo D. Immune evasion of porcine enteric coronaviruses and viral modulation of antiviral innate signaling. *Virus research*, (2016).
2. Lin CM, Saif LJ, Marthaler D, Wang Q. Evolution, antigenicity and pathogenicity of global porcine epidemic diarrhea virus strains. *Virus Res* **226**, 20-39 (2016).
3. Zeng S, *et al.* Proteome analysis of porcine epidemic diarrhea virus (PEDV)-infected Vero cells. *Proteomics* **15**, 1819-1828 (2015).
4. Li W, van Kuppeveld FJM, He Q, Rottier PJM, Bosch BJ. Cellular entry of the porcine epidemic diarrhea virus. *Virus Res* **226**, 117-127 (2016).
5. Farache J, *et al.* Luminal bacteria recruit CD103+ dendritic cells into the intestinal epithelium to sample bacterial antigens for presentation. *Immunity* **38**, 581-595 (2013).
6. RL F, *et al.* 3D visualization of HIV transfer at the virological synapse between dendritic cells and T cells. *Proceedings of the National Academy of Sciences of the United States of America* **107**, 13336-13341 (2010).

REVIEWERS' COMMENTS:

Reviewer #1 (Remarks to the Author):

The authors have answered my questions and have satisfactorily addressed all my concerns. This revision is an improved and much clearer version of the manuscript.

Reviewer #2 (Remarks to the Author):

The revised manuscript fully addresses the previous reviewer comments. It therefore more clearly presents the study design and protocols, as well as the findings, resulting in an improved manuscript that is now suitable for publication.